# Research on the Governance of Rural Living Environments in China: A Perspective of "System-Life" Based on Field Research Conducted in Village A, Xiangtan County, Hunan Province

**Yunjuan Liang [1], Qiyu Shi [1] and Anthony Fuller [2,*]**

1   College of Humanities & Social Development, Northwest A&F University, Xianyang 712100, China; liangyunjuan@nwafu.edu.cn (Y.L.); shiqiyu0529@163.com (Q.S.)
2   School of Environmental Design and Rural Development, University of Guelph, Guelph, ON N1G2W1, Canada
*   Correspondence: tfuller@uoguelph.ca

**Abstract:** This research focuses on the governance of rural living environments in China from the perspective of "System-Life". The objective of improving rural living environments is to construct a beautiful countryside, which is an important part of China's rural revitalization strategy. Through a literature review, a field study, and quantitative analysis, this paper explores the tensions and interactions between local governments and social demand by investigating four elements of the village improvement program: the village's appearance, sewage treatment, domestic garbage disposal, and the sanitation of toilets. We also examine the interactions between the main participants involved in the governance of rural living environments, including the primary-level governments, village committees, and the villagers themselves. It was found that there is a path toward constructing a benign interaction between "system" and "life". In terms of "system", the primary-level governments play a decisive role in the implementation of policies, offering a creative interpretation and flexible implementation of a policy. From the perspective of "life", the village committee is the bridge between the primary-level governments and villagers. The villagers have their own understanding of policy and the logic of life. This probe leads us to suggest that primary-level governments need to respect the perceptions and priorities of villagers in order to improve the performance of this well-intentioned program.

**Keywords:** rural living environments; primary-level government; village committee; governance; system-life; rural revitalization

## 1. Introduction

The improvement of the rural living environments of villages is a key point in China's rural revitalization strategy. Since the CPC Central Committee issued the Three-year Action[1] plan for the improvement of rural living environments in 2018, remarkable results have been achieved in many rural areas. By the end of 2021, with the accomplishment of poverty alleviation, the construction of a beautiful countryside is now one of the development tasks in rural areas.

In China, rural areas in general have been transforming gradually from "dirty and chaotic" to "clean and tidy". For example, the rate of installing sanitary toilets nationwide has exceeded 68%, the proportion of administrative villages[2] that collect and transport household waste has exceeded 90%, and more than 95% of villages carried out the sanitation improvement actions of the plan. The majority of villages are considered to be clean, tidy, and orderly[3]. With the progress of urban–rural integration, the rural environment has also become a place for urban residents to fulfill their desire for a simple and traditional country experience [1]. These actions have not only improved village living conditions but also brought satisfaction, a sense of fulfillment and respect, to the residents [2]. Previous studies

show that the improvement of rural living environments can improve public services and infrastructures in poorer areas, as well as lift the spirits of those in poor rural households, which is a way of improving the effectiveness of poverty alleviation [3]. The reciprocal relationship between the rural living environment and rural tourism is also a good starting point for rural revitalization [4]. In the process of environmental improvement, the formal system and the willingness of villagers must function synchronously to reshape public values and promote better community living.

However, in some rural areas, the governance of living environments is not without problems. For example, it is often the case that an effective coordination mechanism has often not been established between different agencies. This raises the following important questions: To what extent has village people's environmental protection awareness increased? When their participation is limited, how can villagers build an efficient and effective interaction mechanism for all participants? Effective local governance is still a difficult problem in much of rural China today.

We chose the perspective of "System-Life" rather than "State-Society" to analyze the dynamic between institutions and individuals at the micro level by focusing on the fundamental component of the rural governance system: village people. There is a certain degree of differentiation between the system and its agents; this consists of the primary-level governments[4] and the villagers' interest in and willingness to cooperate. There is also tension in the relationship between village committees[5] and villagers. The village committee is the executor of policies in rural areas. It is also responsible for collecting feedback from villagers. In addition, the vague boundaries of "system" agents lead to uncertainty during the implementation of policies, which interferes with interaction mechanisms and tends to lower the participation rates of villagers.

The interaction between system and life is designed to undergo five stages: system design, system introduction, system operation, feedback from the subjects, and system updating [5]. The central government is the general designer of the overall system. The governance of rural living environments needs to be implemented through the creative actions of institutional agents. Thus, the "last mile" of the implementation of a policy can be executed by the village committee; its function is like the bridge between the system and life.

As the values and interests of villagers are becoming increasingly diverse and complex, interaction between local governments and villagers is also becoming more frequent. This leads to the following questions probed in this research: (1) How do environmental improvement policies impact rural living environments according to the village residents? (2) How do we define the role of the main actors, including local governments, especially primary-level governments; village committees; and the villagers themselves, in the governance of rural living environments? (3) How do we respect and include the rules and mores of villagers so that their greater participation in the process of rural governance is effective?

## 2. Literature Review

"State-Society" is the traditional perspective used in domestic and international academic circles when studying social change in China [6]. In the 1990s, Joel S. Migdal proposed the paradigm of the State in society. "State" and "Society", as two opposing entities, were shown to constitute a broad yet feasible perspective [7]. The "State-Society" perspective corresponds to the reality of increasing differentiation and opposition in social classes after the Industrial Revolution [8]. It presupposes two basic categories, namely, "State" and "Society", both of which are systems with their own logic and boundaries. During research, the relationship between these two manifests as a trade-off interaction, such as "small government, big society" or "strong country and weak society". The primary-level governments are the forefront connection between the State and Society. The governance of rural living environments is an important part of primary-level governance, which reflects the dual aspects of the State and Society [9].

Existing studies have conducted in-depth discussions about the governance of rural living environments based on this traditional perspective of "State-Society". Primary-level governance is the foundation of central governance [10]. The perspective of central governance emphasizes the technocratic dominance of the State over all levels of government, focusing on the impact of top-down central governance on rural living environments [11]. Resources are allocated to the primary-level governments, and corresponding institutional arrangements can be made. The primary-level governments need to formulate flexible policies and mechanisms, such as ideological guidance, demonstrations, and measures adaptable to local conditions, through grassroots organizations such as village committees. Then, the "hard rules" of the State can be integrated into the lives of the villagers [12].

The interaction between the "State" and "Society" is manifested within the hierarchical administrative system, of which different levels of government take measures to fulfill administrative tasks designed to improve the living environment in rural areas. These measures include refining relevant legislation [13], implementing environmental governance policies [14], utilizing digital governance platforms [15], and employing modern governance technologies such as information networks [5].

Based on relevant theories such as collaborative governance[6], Jiang Lina et al. discussed the influence of policy and the institutional environment on villagers' willingness to classify domestic waste and its removal [14]. Liu Peng et al. proposed, from the perspective of the legal system, that the construction of rural living environments should follow the rule of law [13]. In general, the existing studies show that institutions have a positive effect on the governance of rural living environments in terms of resource input, institutional guarantees, and administrative supervision, reflecting the active intervention of all levels of government in the governance of rural living environments. However, there are also problems, such as "one-size-fits-all" policy arrangements, which lack adaptable measures for local conditions [16], resulting in high resource input, low efficiency, and high governance costs [17]. It is suggested that governments should increase resource investment, upgrade resource utilization methods, and build a long-term, standardized, and institutionalized path for the governance of rural living environments [18]. State-led governance usually adopts "top-down" and "project-based governance" schemes. At the same time, local governments also use more tactical methods for specific policy implementation [15], hampering the ability of the primary-level governance model followed by the State to be adaptable to rural social life [14]. Improving rural areas where public spirit has declined may not be enough to solve embedded problems.

Some studies point out that the State and Society are not enemies, and determining how to effectively build a relationship between them has become a key point in the debate on the governance of rural living environments. The combination of administration and autonomy should be explored, and governance models such as "collaborative governance" and "embedded governance" should be further tested to improve the performance of rural living environments and provide better living conditions for the villagers [18].

These studies mainly focus on resource allocation, institutional guarantees, and institutional empowerment, emphasizing the tension and reconciliation between the State and Society. Under the framework of the "State-Society" formulation, there are many contradictions and ambiguous areas regarding the governance of rural living environments, such as the mismatch of national goals and social needs. Some models combine administrative leadership and autonomy, autonomy that emphasizes the participation of villagers and other social organizations [9]. However, these studies also note problems with the governance behavior itself, which often neglects the logic of villagers' daily lives. With the development of rural areas, many interests are increasingly differentiating, and social life has become more diverse and complex. Chinese scholar Xiao Ying has proposed that "system" refers to the formal system created by governments and implemented by agents at all levels. It is based on clear values, theory, and rules. Simplification and clarity are its main characteristics. "Life" refers to people's daily activities, which are practical, with fuzzy boundaries, routines, toughness, and resilience [5]. There is also tension between

"system" and "life". The subject of life corresponds to the villagers, who are the specific bearers and constructors of their lives. They have their own logic and demands, as well as the ability to create the corresponding knowledge according to their situations. Xiao Ying points out that the interactive practice of "System-Life" could be used to analyze the changes in and mechanisms of "public opinion" in the formal system.

The "System-Life" perspective incorporates the subjectivity of the villagers and social organizations in rural governance, and the system plays a supportive role in activating the potential of the villagers. As Figure 1 shows, the initiative of villagers is an important part of effective governance and policy implementation in rural areas. In calling for the enthusiasm of villagers' participation to be stimulated, Xiao Ying writes that "the efficiency and the supply of rural public goods should be improved, and good governance should be promoted by autonomy" [19].

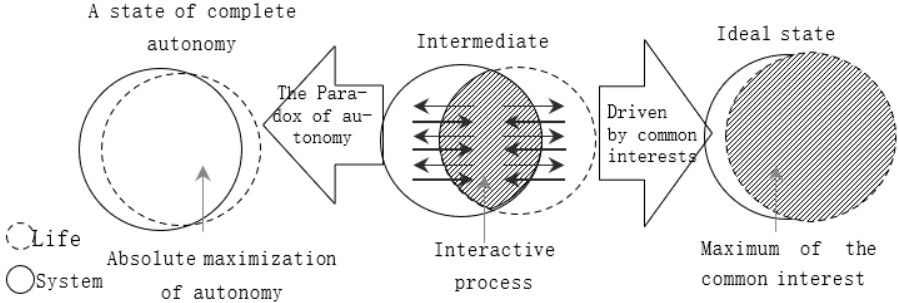

**Figure 1.** Interaction mechanism of "System-Life".

Villagers are victims of environmental damage and beneficiaries of environmental improvements; they are the participants in environmental remediation and the subjects of environmental policies, all of which determine villagers' active and sometimes dominant role in rural governance [20]. Under the current dramatic changes in China's social structure, with the rapid development of the market economy, the differentiation of rural classes has intensified, and social values have been diversified [12]. The public spirit of villagers is diminishing, and the role of village people is gradually weakening. The literature suggests that interest incentives and institutional empowerment are important for reshaping the fundamental role of village people in the governance of living environments. In the context of governance modernization, almost all villagers are encouraged to participate in governance through specific institutional practices such as "Ethical Banks"[7] and "Points Systems"[8]. In the process of implementing the policy under the administrative bureaucracy system, however, there is a lack of consideration of the main role of villagers. In addition, the institutional arrangement lacks both specificity and universality [21].

In Western countries, after the Second World War, research was initiated to critically examine the impact of urbanization on rural environments. Charron and her colleagues investigated the application of a novel governance technique known as the "Integrated Planning Scorecard" in the governance of a rural community in the USA, highlighting its effectiveness in involving various stakeholders in rural environmental enhancement efforts [22]. The "Integrated Planning Scorecard" is similar to the "Points Systems" practice used in modern rural China. Japanese scholars Tabimoto K and Mori K believed that local governments, in governing rural environments, should take into consideration aspects of the living environment that significantly impact residents' settlement decisions. This includes providing employment opportunities and ensuring transportation and healthcare conditions. Local governments are advised to implement governance measures based on residents' demonstrated needs in these areas [23].

In the current period, research on rural environmental issues has primarily focused on the interplay between human and natural environments. The key issues here include rural environmental pollution, rural community transformation [23], and a growing emphasis on specific rural environmental concerns such as waste disposal, domestic sewage treatment,

and landscape revitalization [24]. For instance, Mihayi et al. examined the problems of the illegal dumping of waste in rural Romania, revealing significant environmental damage caused by vast quantities of discarded waste and inefficient collection measures in rural areas [25]. In the domain of rural sewage treatment, Cooper and his team researched the preferences of Australian rural villagers for improving domestic sewage treatment, finding that villagers were willing to pay for domestic sewage treatment to enhance their living conditions [26].

As shown in Figure 1 above, there is an interactive process between "System" and "Life". It is driven by the common interest of local governments, village authorities, and villagers. If the common interest were to be maximized, it would need to undergo an interactive process to gain a semblance of autonomy. In reality, the complete autonomy of villagers cannot be realized due to resource limitations, the capacities of village people, and the lack of public spirit.

## 3. Methods

In this paper, we use an in-depth literature review to trace the experience regarding the governance of rural living environments from the "System-Life" perspective in comparison to the more familiar State–Society view. We use a village case study and quantitative methods to evaluate the performance and satisfaction of the villagers with respect to the policies and changes in their living environments. We also examine the interactions of "System" and "Life", exploring the implementation of the "Three-year Action" plan to improve landscape appearance, garbage disposal, sewage treatment, and toilet renovation.

In the survey, all the villagers from village A, Xiangtan County, were counted as a whole, and the survey subjects were selected via cluster sampling according to centralized contiguous living arrangements and village groups. Data were collected using a questionnaire, participatory observation, in-depth interviews, and overall case analysis. A total of 315 questionnaires were distributed, of which 302 were valid upon being returned, with an effective rate of 95.9%. Most of the invalid questionnaires were submitted by elderly villagers over 70 who had difficulty answering some of the questions. We also visited the relevant departments of the government of Township S. Eleven staff who were in charge of environmental governance were interviewed.

### 3.1. Research Subject and Data Collection

We chose village A as the research site; the reasons for this choice are as follows. First, village A is located in the central region of China, dominated by a small-scale peasant economy. Villagers' daily lives, beliefs, and decisions are less affected by external impacts, and many traditional customs are maintained. Second, since 2018, the government of Township S has promulgated a series of policies related to the governance of rural living environments, and the response of villagers in village A was more active than that of other villages. Thus, we could look into policy implementation at the "system" level, as well as the villagers' demands from the aspect of "life". Third, during and after the Three-year Action plan's implementation, the living environments of village A were significantly improved, and it has been regarded as a model village; its benefits were publicized from time to time by Township S. These reasons for the selection of village A comprehensively reflect the interactions of different actors in the governance of living environments from the perspective of "System-Life".

The interviews were conducted with 302 villagers. The questionnaire survey included six sections: basic household information, villagers' attitudes towards their village's appearance, domestic sewage, household garbage, toilet renovation, and contact information of correspondents. A total of 113 variables were devised. The main purpose of the interviews was to measure the villagers' recognition of and satisfaction with the governance of their rural living environment and their expectations for governance in the future. The statistics on gender, age, employment, education, and political identity of the 302 valid samples are shown on the Table 1 below.

**Table 1.** Sample distribution in village A.

| Gender | Ratio | Sample | Education | Ratio | Sample |
|---|---|---|---|---|---|
| Male | 53.60% | 162 | Uneducated | 5.30% | 16 |
| Female | 46.40% | 140 | Primary school | 31.80% | 96 |
| **Age** | **Ratio** | **Sample** | Junior high school | 38.70% | 117 |
| 18 years old and below | 4.00% | 12 | High school or vocational high school | 10.60% | 32 |
| 19–35 years old | 25.50% | 77 | Technical secondary school or technical school | 1.30% | 4 |
| 36–70 years old | 45.40% | 137 | Junior College | 8.30% | 25 |
| Over 70 | 22.50% | 68 | Bachelor degree and above | 7.90% | 24 |
| **Employment** | **Ratio** | **Sample** | **Political identity** | **Ratio** | **Sample** |
| Farming | 53.00% | 160 | Members of Communist Party | 6.00% | 18 |
| Farming-dominated and non-agricultural sector | 11.30% | 34 | Members of Communist Youth League | 11.9% | 36 |
| Non-farming business | 15.90% | 48 | | | |
| Non-agricultural and agricultural farming mainly | 4.00% | 12 | No political affiliation | 74.2 | 224 |
| Out of work | 5.30% | 16 | | | |

### 3.2. Pretest of the Dat

In this study, statistical data analysis was conducted for the valid questionnaires. IBM SPSS v26.0 software was used to collect data for coding and analysis. According to the purpose of the study, a two-stage statistical analysis of the pretest and formal questionnaires was conducted. The pretest focused on the reliability and decision value analysis of the questionnaire. The statistical analysis in the pretest part mainly included factor analysis, Cronbach's $\alpha$ value, and decision value analysis. As for the statistical analysis of the formal questionnaire, statistical analyses of narratives, factor analysis, validity analysis, and reliability checks were conducted.

The KMO test is a measure of sampling adequacy proposed by Kaiser, Meyer, and Olkin. The KMO test tests the relative magnitude of the simple and partial correlation coefficients between the original variables. The calculation formula is

$$\text{KMO} = \frac{\sum\sum_{i \neq j} r_{ij}^2}{\sum\sum_{i \neq j} r_{ij}^2 + \sum\sum_{i \neq j} r_{ij \cdot 1,2\ldots, k}^2}$$

Commonly employed KMO metrics were used for validity analysis: a score of 0.9 or more means very suitable, 0.8 means suitable, 0.7 means average, 0.6 indicates not very suitable, and below 0.5 indicates highly unsuitable. The KMO statistic ranges between 0 and 1. When the sum of squares of the simple correlation coefficients between all variables is much greater than the sum of squares of the partial correlation coefficients, the KMO value is close to 1. The closer the KMO value is to 1, the stronger the correlation between the variables, and the more suitable the original variables are for factor analysis. Oppositely, the closer the KMO value is to 0, the weaker the correlation between the variables and the more unsuitable the original variables are for factor analysis.

According to the results of the exploratory factor analysis shown in Table 2, the coefficient result of the KMO test was 0.789, showing that the validity of the questionnaire is good. According to the significance of the spherical test, it can also be seen that the significance is close to 0. Therefore, the questionnaire has very good validity.

**Table 2.** KMO and Bartlett tests.

| Number of KMO Sampling Suitability Quantities | | 0.789 |
|---|---|---|
| Bartlett spherical test | Approximate chi square | 3670.460 |
| | free degree | 351 |
| | significance | 0.000 |

**Reliability analysis:** Cronbach's $\alpha$ is the most widely used reliability index. According to the value of the item deleted of Cronbach's $\alpha$ in the consistency test, it is recommended to eliminate the highest value of each item. As Table A1 shows in Appendix A, the standardized reliability coefficient of each part of the questionnaire is 0.864, and the reliability coefficient ranges from 0 to 1. The closer it is to 1, the higher the reliability. The result of this analysis was 0.864, indicating that the overall confidence of the questionnaire is very high. The test process was implemented using exploratory factor analysis. According to the deleted reliability coefficient, the value was less than 0.864 on the whole. Therefore, the questionnaire for examining rural living environments does not need to be adjusted.

## 4. Results: General Situation of the Governance of Rural Living Environments in Village A

Village A is located in Township S, in the west of Xiangtan County, Hunan Province, China. In the first quarter of 2019, village A was listed as one of the "five bad villages"[9], while in the fourth quarter of the same year, it was ranked among the "five good villages". Moving from "bad" to "good" in township S is indicative of a significant change in the quality of its ecological environment, especially in terms of village appearance, domestic sewage, garbage disposal, and toilet renovation.

### 4.1. Village Appearance

The appearance of the village not only reflects the image of the countryside and the living environment of the villagers but also its ability to attract rural industry and external investment prospects. Since January 2021, village A has been carrying out roadside sanitation patrols and door-to-door household sanitation inspections. In March 2021, village A signed a contract with the company that was in charge of village A's sanitation; the contract included garbage transfer, daily cleaning, and household garbage collection, with an annual payment of CNY 68,000 (USD 9510) from the village committee. The contract specifies the reward and liability for a breach. For example, if the sanitation of village A is rated as "First-level" in the evaluation of the upper-tier level of the government, the company can be rewarded CNY 3000 (USD 420). For "Second-level" and "Third-level", the rewards are CNY 2000 (USD 280) and CNY 1000 (USD 140), respectively. A "Fourth-level" designation would receive neither a reward nor punishment. If the result is "Fifth-level", the company would be punished, including by being criticized by superiors, complained about by the villagers, and fined CNY 200 (USD 30) to CNY 1000, and they may even have their contract terminated. The village committee transformed from being the executor to the supervisor of sanitary improvement by purchasing some social services to reduce its administrative work.

With the above measures implemented, the villagers were satisfied with the appearance of their villages. Seventy-seven percent of the respondents maintained that the village's appearance was good, and 66% believed that the levels of greening and environmental protection were relatively good. However, the field survey shows that the villagers were not satisfied with the water quality of the ponds and rivers as well as the irrigation system. Almost a quarter of the respondents pointed out that the water in ponds and rivers was not clean, and 35.8% of them thought that the irrigation system was neither convenient nor sufficient.

*4.2. Domestic Sewage*

To draw water for domestic use in village A, the villagers still depend on well water; only 11.7% of the villagers have tap water facilities installed (see Table 3). According to the responses in the in-depth survey, the village cadres were not willing to promote tap water installation either; additionally, water is cheaper than tap water. Meanwhile, some areas cannot connect to the tap water system due to difficult terrain. However, the most important reason reported is that most villagers prefer to use well water. When we look at the villagers' daily lives, it seems they are not too concerned about water safety, as long as the water looks clean and has no particular smell. Without administrative measures, the wells in the village may exist for a long time to come.

**Table 3.** Domestic water intake in village A.

|  | Water Intake | Percentage |
|---|---|---|
| **Household Water Intake** | Tap water | 11.7% |
|  | Well water | 86.9% |
|  | Other | 1.3% |
|  | Total | 100.0% |

Water intake, water use, and drainage are the three major steps in domestic water use, and they are also important considerations in the construction of a good living environment. Arguably, villagers ought to pay great attention to the treatment and usage of water resources since sewage can affect their health and the quality of agricultural products directly. Nevertheless, most of the villagers neglected this consideration. Thirty-seven percent of the respondents chose to pour used water directly on the ground, and forty-nine percent preferred to discharge it into the sewer, while sixty percent of the respondents believed that the pond sewage treatment did not reach a "good" standard. Only 33% knew about the environmental harm of sewage. Seventy-three percent of the respondents did not think that governments should be involved in the process of sewage treatment. They generally believed that domestic sewage had little impact on their living environment and that the existing ditch drainage system is sufficient.

Villager H (60 years old) stated that "the sewage finally flowed from this small channel to the river. We use it to irrigate the land directly...I do not think the village is capable of dealing with the sewage. However, the sewage seems not harmful to us" (Figure 2).

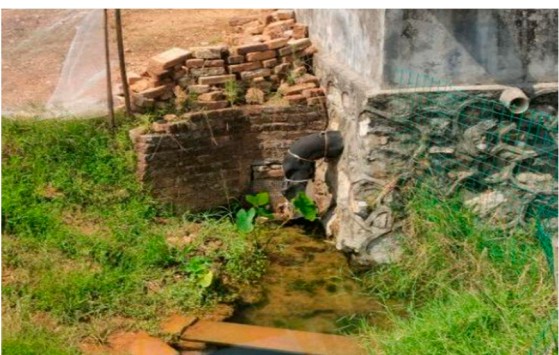

**Figure 2.** Household sewage is directly discharged into the adjacent river.

Some villagers maintained that the primary-level government was reluctant to become involved in sewage treatment because it costs too much. However, given the field research, we found that most of the villagers were content with their recent life status, and they were used to their original way of living. In recent years, village A has improved its sewage discharge by repairing and dredging channels, as well as by digging deep channels with the support of the primary-level government of Township S. Sewage can now be discharged

into a septic tank instead of into the ground. However, fewer than 20% of respondents used a septic tank to deal with sewage.

### 4.3. Garbage Disposal

In addition to having garbage cleaning and transfer outsourced to the agreed company by the village committee, "two-color" garbage cans for "recyclable garbage" and "other garbage" were purchased and placed in front of each house by the village committee. Garbage disposal has always been the key point for improving the living environment in the village. No less than 70% of the villagers had started simple garbage classification.

Villager L said that "environment protection should involve every villager. Garbage classification is indeed good for the village. Meanwhile the garbage that used to be burnt is still burnt. For example, in the past, the fallen leaves were usually burnt. Nowadays this kind of treatment of fallen leaves was forbidden as it may pollute the air and it is also unsafe. Yet most people still prefer to burn and then bury them, which makes good fertilizer...plastic bags are sometimes burnt directly too" (Villager L, 70 years old).

From the statement of villager L, it can be gleaned that most villagers have their own ways of disposing of garbage, such as burning, although this is forbidden. Some villagers were ignorant of the fact that plastic should not be burned because harmful gases can be released.

Garbage reduction and garbage recycling are performed based on garbage classification. However, through field survey questionnaires, it was found that rural garbage classification and recycling was not very effective. Less than 60% of the villagers believed that the effect of garbage classification on the village was good, even though it is a very important part of garbage disposal. The problem for both the village committee and the villagers is that it is more difficult to implement garbage classification in rural areas. Therefore, in the process of policy implementation, "garbage classification" may not be as important as "waste reduction".

In the questionnaire section measuring villagers' knowledge of garbage classification, almost all the participants failed except the youths who had been educated. The villagers were reluctant to accept the relevant knowledge on garbage classification. They believed that the primary-level government did not convey garbage classification policies and requirements very well; hence, they neglected this practice to some extent. Additionally, the villagers had their own ideas and logic for dealing with garbage. It seems they do not care if it they are employing a wrong way of dealing with garbage or if it is harmful to their health and environment. What they are concerned about is if garbage disposal is convenient for them or if they can benefit from it. Carrying out garbage classification has been challenging in urban China, let alone in rural areas. This shows that the impact of the policies on rural life regarding garbage classification is limited.

Villager M stated that "there are two different colors of garbage cans in front of my house, only one garbage truck comes to collect garbage every week, and the garbage in both cans would be poured into the truck together, there is no sorting at all. Suppose using two garbage trucks is not cost-effective (Villager M, 58 years old)".

There seems to be a delicate relationship between the system and the villagers' lives, with most villagers accepting the payment of CNY 30 (USD 4.2) per household per year for garbage disposal using garbage cans, even if they have doubts about the cost. In daily life, a convention of garbage disposal has been formed. That is, the villagers have the right to deal with kitchen waste and dead leaves via composting and reuse by their own means and freely. Other forms of large plastic packaging and waste are required to be put into garbage cans. Therefore, it can be considered that the refinement of the garbage system is the foundation of the development of improved living conditions in village A. Traditional habits have changed under the norms of the system, and the required changes are being understood and accepted by the villagers in different ways.

From the measures shown above, we can see that there is a compromise between the System and Life, and the village committee is an important bridge in this interactive

relationship. The "last mile" of policy implementation depends on the village committee. In the "System-Life" perspective, the goals of the policies and the interests of the villagers may have a certain degree of difference. There is still some tension between them. While the village committee is the intermediary between primary-level government and the villagers, it should not only optimize and implement policies for improving rural living environments but also adjust policies according to the feedback from the villagers. Finally, the village committee is seeking practical ways to conform to the opinions of the villagers and the reality of village life in rural China.

### 4.4. Toilet Upgrade [10]

Nearly half of the villagers in village A renovated their houses by themselves before the "toilet revolution". Due to limited funds, village A was not allocated the resources for toilet renovation until 2021. The toilet revolution started with the administrative order of the primary-level government. To implement the task, the village committee promoted it widely among the villagers but to little effect. Less than half of the villagers said that they had not heard of the "toilet revolution" provision. Among the 64.2% of villagers who installed septic tanks in the past, most did not think it was necessary to renovate their toilets again. Only 10% of them chose to implement toilet renovation in 2021. Despite the subsidies (CNY 200) provided by the primary-level government of Township S for toilet renovation for each household, the villagers were still reluctant to renovate their toilet as the subsidy was not enough.

The resistance to the "toilet upgrade" in village A was not strong. At the time of writing this manuscript, the villagers could still voluntarily renovate their toilets by themselves with a subsidy of CNY 300 (USD 42) per household provided by the township government. Before March 2021, there was a bank of 100 places for the villagers to apply for a toilet renovation. By the end of 2021, almost eighty places had been taken up. That is to say, most of the villagers renovated their toilets, even though the subsidy was not enough.

It can be found from Table 4 that many more households used water closets than dry closets, which demonstrates that the benefits of sanitation brought by the toilet upgrade satisfied the villagers. In the beginning, the villagers exhibited a "wait-and-see" attitude towards the toilet upgrade provision, either because the subsidy provided by the local governments was insufficient or because they did not see the benefits of renovating the toilets. Eventually, they were convinced by peer pressure and the regulations of the township governments.

**Table 4.** Current household toilets in use in Village A.

| Dry Closet | The Toilets Currently in Use | | | Total |
| | Water Closet | Both | Refuse to Answer | |
| --- | --- | --- | --- | --- |
| 28 | 0 | 4 | 4 | 36 |
| 20 | 206 | 12 | 0 | 238 |
| 4 | 16 | 8 | 0 | 28 |
| 52 | 222 | 24 | 4 | 302 |

Most villagers who renovated their toilets originally held the opinion that "the elderly were not used to water closets, so they do not want to change them". However, in Table 5, it is shown that the respondents at any age were willing to choose water toilets, even the elderly above 70; the number of villagers choosing water toilets was far greater than that choosing dry toilets.

**Table 5.** Villagers' toilet preference arranged by age of respondents.

| | | Which One Is More Practical? A Dry Toilet or a Water Toilet? | | | Total |
|---|---|---|---|---|---|
| | | Dry Toilet | Water Toilet | All Practical | |
| | 70 and above | 16 | 73 | 4 | 93 |
| | 55–70 | 10 | 69 | 2 | 81 |
| Age of all respondents | 35–55 | 6 | 68 | 2 | 76 |
| | 19–35 | 0 | 20 | 20 | 40 |
| | 18 and below | 4 | 8 | 0 | 12 |
| | Total | 36 | 238 | 28 | 302 |

It can be seen from Tables 5 and 6 that almost all the villagers, including the elderly, preferred the water toilets, and most villagers with primary-school education and above held a more positive view of water toilets. This result demonstrates that the willingness to renovate toilets is not directly related to age but education.

**Table 6.** Villagers' willingness and education levels.

| | | Which One Is More Practical: A Dry Toilet or a Water Toilet? | | | Total |
|---|---|---|---|---|---|
| | | Dry Toilet | Water Toilet | Both Practical | |
| | Without any education | 8 | 8 | 0 | 16 |
| | Primary school | 16 | 74 | 0 | 90 |
| Educational levels of all all respondents | Junior middle school | 2 | 106 | 12 | 120 |
| | Senior high school | 0 | 52 | 0 | 52 |
| | Technical secondary school or technical school | 1 | 7 | 0 | 8 |
| | College or University | 1 | 8 | 2 | 11 |
| | Other | 4 | 0 | 0 | 4 |
| Total | | 34 | 217 | 14 | 302 |

*4.5. The Interactive Mechanism between "System" and "Life" in the Governance of Rural Living Environments*

In the surveys, we investigated the villagers' attitudes towards "village appearance", "sewage treatment", "garbage classification", and the "sanitation of toilets". "I am willing to participate in the activities to promote the village appearance" was the main response of the villagers towards environmental improvement. These items each have correlations above 90%, which shows that the villagers with a greater desire for environmental improvement were more satisfied with rural environmental governance. If the villagers' awareness of good living environments is improved, they will directly or indirectly invest in the construction of further improvements.

It can be seen from Table 7 that there is a significant correlation between the five variables and that the correlation coefficient is greater than 0, which is a positive result. This shows that the contradiction of the villagers' response to the policies is also reconcilable.

Garbage classification is used to recycle household garbage in cities. It requires specific knowledge and good habits. Long-term practices have shown that it is extremely difficult to promote and implement garbage classification in some urban areas, let alone in rural areas with generally low education levels and a more traditional lifestyle. The villagers' polarized choice of "two-color" garbage cans shows that some of them lack knowledge of garbage classification, which also indirectly reflects the shortage of corresponding local resources. When governments with a technocratic "system" do not understand the actual situation regarding "life subjects"—the villagers—the logic of the institution will probably contradict the logic of life. Nevertheless, this contradiction is not irreconcilable.

**Table 7.** Correlation analysis of villagers' satisfaction with environmental governance.

| Variables | Correlation | Participation in the Improvement of the Village's Appearance | The Village's Appearance Is Good | Satisfaction with Domestic Sewage Treatment | The Effect of Garbage Classification Is Good | Satisfaction with the Sanitation of Toilets |
|---|---|---|---|---|---|---|
| Participation in the improvement of village appearance. | | 1 | | | | |
| The village's appearance is good. | Pearson correlation | 0.151 ** | 1 | | | |
| Satisfaction with domestic sewage treatment. | | 0.028 | 0.094 | 1 | | |
| The effect of garbage classification is good. | | 0.149 ** | 0.093 | 0.037 | 1 | |
| Satisfaction with the sanitation of toilets. | | 0.383 ** | 0.310 ** | −0.038 | 0.233 ** | 1 |

At the 0.05 level (two-tailed), the correlation is significant. **. At the 0.01 level (two-tailed), the correlation is significant.

According to the results of multiple comparisons, Table 8 shows that the villagers that were more or less satisfied with the environmental policies showed stronger support for environmental protection than those with an indifferent attitude (I "do not care"). The villagers with good environmental awareness tend to have more knowledge related to environmental protection. Although some rural policies are not eco-friendly or reasonable, this does not mean that every policy has problems. Therefore, the contradictions between the villagers' willingness and environmental policies are not irreconcilable.

**Table 8.** Differential analysis of villagers' satisfaction with the environmental policies of village A.

| Variable | Satisfactory | Number of Cases | Average | Standard Deviations | F | Sig | Multiple Comparisons |
|---|---|---|---|---|---|---|---|
| Attitude towards the "two-color" garbage cans | Do not care | 34 | 2.41 | 0.844 | 3.471 | 0.033 | 2 > 1, 3 > 1 |
| | Care less | 100 | 2.7 | 0.631 | | | |
| | Care more | 16 | 2.46 | 0.761 | | | |
| Domestic sewage treatment is the responsibility of the governments | Do not care | 34 | 2.45 | 0.73 | 1.077 | 0.343 | / |
| | Care less | 100 | 2.39 | 0.731 | | | |
| | Care more | 16 | 2.19 | 0.801 | | | |
| The publicity of village appearance remediation is necessary | Do not care | 34 | 2.34 | 0.776 | 0.253 | 0.777 | / |
| | Care less | 100 | 2.42 | 0.644 | | | |
| | Care more | 16 | 2.35 | 0.745 | | | |

Note: 1 represents "do not care", 2 represents care "less", and 3 represents care "more".

With the development of the economy and the different interests in the village, the participants in the framework of "State-Society" are becoming increasingly diverse, complex, and problematic from the perspective of traditional structuralism. In contrast, the "System-Life" approach can be used to analyze the relationship between the system and life in a micro and dynamic way. It can also be used to investigate the conflict and interaction mechanism among the participants. Due to the mismatch between the policy of improving living environments and daily life from the logic of governance, there are some difficulties in the current governance approach to improving living environments. Specifically, the policies are usually implemented via a "one-size-fits-all" approach without considering the different conditions in the villages. Moreover, the implementation of the policy is usually led by a "Demonstration Village" or "Star Village", which means various increased resources are allocated by the primary-level government. Such policies can hardly apply to the disorderly conditions of ordinary villages. The most important aspect is to understand

the needs and conditions of different villages and apply appropriate policies and resources to them, balancing the villagers' interests and the policy imperatives.

Although there is an inevitable tension between the "system" and "life", in reality, the system is largely constructed by life. Although the system cannot completely dominate life, it can change life to some extent. If the system is to serve life better, it is necessary to resolve these conflicts to promote benign interaction between the two and render the system better able to respond to the varying conditions of rural life.

## 5. Discussion

### 5.1. Low Participation of Villagers in Rural Governance Is the Greatest Issue

Villagers constitute the main body in the governance of rural living environments, and they are also the direct beneficiaries of any environmental improvements. The governance of rural areas clearly requires more participatory efforts from villagers as the current level of participation is not effective. For example, village A did not encourage villagers' participation in the act of rural governance, as the village committee signed the contract directly with the company that was in charge of village A's sanitation, including daily cleaning, household garbage collection, etc. However, the village people needed to pay for this. The effect of sanitation is positive, but it is not good in terms of public awareness and willing participation. Some village people held the opinion that "it is natural to let the staff of the company do the cleaning things, because I have paid for it." In that case, village people will gradually pay less attention to village sanitation. There are some good experiences regarding rural governance in other villages in rural China, and much can be learned from them. For example, the village committees of other villages hired some residents, usually older people who are retired and in good health, to carry out the daily cleaning of the village. This approach makes better use of rural surplus labor, while the participants become more engaged in environmental issues.

In the field study of the "toilet revolution", it was found that almost half of the respondents did not know anything about it, so they did not participate. To address this situation, the village committee needs to strengthen public awareness and, if necessary, visit households to explain the regulations and opportunities.

In addition, through the survey, we found that most of the villagers preferred water toilets, but that the subsidy per household for "toilet revolution" was not sufficient, which reduced enthusiasm for village participation. To promote the "toilet revolution", we can rely on more than just administrative orders; more importantly, given the adequate subsidies available, we can also pay attention to the villagers' demands. If they can afford to renovate their toilets, most would like to participate in the upgrade process.

### 5.2. Rigid Implementation of Policies, Neglect of Rural Practices, and the Willingness of Villagers

Rural living environments have both public attributes such as community environmental resources and private attributes such as the living conditions of villagers. The policies need to be implemented flexibly in order to respect the realities of village life. However, some policies were implemented in village A rigidly; for example, the village committee purchased "two-color" garbage bins for each household to enforce garbage sorting while asking the villagers to pay for it. This occurred because it was a task allocated by the primary-level (Township) government. Although the cost was not high, it did not respect the villagers' willingness or their situation. The effect of garbage classification was not good either. In this situation, the governance of garbage disposal has become a formality, and public resources are not being effectively directed toward the well-being of the villagers.

Interests, vested or otherwise, are always at play in the social field. The benign interaction between "system" and "life" in rural governance aims to build a benefit-sharing relationship between institutional objectives and life subjects. Thus, governments need to understand the most important and practical needs of the villagers so as to allocate resources to the issues that are closely related to their lives, such as the improvement of water irrigation facilities on farmland and road maintenance. For the villagers, it is necessary

to express their actual demands to the village committee and primary-level government clearly. Where possible, the complexity of village demands needs to be respected.

## 6. Conclusions

According to the perspective of "System-Life", the interaction between system and life must undergo five stages: system design, system introduction, system operation, feedback of the subjects, and system renewal. As this paper shows, rural residents are the main body involved in rural governance but also the direct beneficiaries of the improvements of rural living environments. Therefore, system design should respect the opinions of village people. However, in practice, decisions still mainly depend on the level of government decision making, especially those in which the villagers' participation is not active. In some villages with poor environmental conditions, the awareness of village participation is even weaker. This makes the system renewal stage remain in the step of "system operation", which lacks feedback and prevents the system from being updated.

In the governance of rural living environments, the primary-level government usually solves problems through mandatory measures, such as administrative orders and regulations. To achieve a short-term effect, the policies are always of a "one-size-fits-all" nature and as such unable to achieve the goal of governance, but they may activate the contradiction between system and life, even causing unnecessary disputes. Meanwhile, without fully considering the villagers' demands and willingness, although there are long-term policies, the lack of a set of supervision and scientific assessment mechanisms may lead to the result of being "locally effective, overall slow". The governance of rural living environments is a whole system; it requires coordination between all subjects.

In terms of "system", the primary-level governments, which play a decisive role in policy implementation, must establish interdepartmental contacts to realize orderly governance, promote system construction, and increase technological investment so as to provide a strong guarantee for improving the comprehensive management of rural areas. In the preparation of rural planning, the villagers' ideas should be respected, considering the differences in life and levels of autonomy, building on local characteristics, and, with long-term planning, establishing a sustainable management and protection mechanism. In this regard, local officials might need more training in working with small community groups to promote the benefits of environmental improvements. In order to ensure the comprehensive management of living environments, the government should better understand the needs of rural residents, clarify their priorities, and implement step-by-step plans according to the resource differences in adjacent regions. In this way, local governments can establish a long-term management guarantee mechanism and include a political and financial guarantee. At the same time, it can establish a social supervision system in rural areas to carry out a watching brief over the relevant government departments. On the basis of clarifying the responsibilities of all parties, comprehensive cooperation should be conducted.

From the perspective of "life", village people have their own understanding of a policy. It is impractical to use administrative means to enforce policies in most Chinese rural areas. The improvements in living environments could increase the happiness of residents, their sense of progress, and community development. This is the key to safeguarding the vital interests of these people in the future.

Regarding rural governance, it is becoming increasingly important to give full play to the organizational functions of village committees, which are the intermediaries between the local government and the villagers. Village committees must come to understand the specific situation of their villages and the needs and assets of the villagers. Accordingly, they would be of invaluable assistance to the implementation of rural policies.

There is usually a tension between the "system" and "life". The system cannot regulate life completely. On one hand, life has its own logic and self-organizing ability, which might seem not only noisy and disorderly but also creative and dynamic. On the other hand, a high-standard and strict evaluation system consumes many resources [27], but, as shown in this study, such a system has not improved the lives of villagers very much. Restricting

system expansion and leaving space for the living subjects to coordinate themselves are important. It is critical to respect the regulations and laws of rural society and to establish scientific and reasonable governance goals. The primary-level governments need to use diversified policy implementation methods to support the positive role of grassroots organizations. In terms of implementation, the system needs to be more flexible.

The observations derived from the objective survey are conditioned by the limitations of village A being a case study. They are strengthened, however, with reference to the literature, which collectively reveals similar results for many other environments in rural China.

**Author Contributions:** Conceptualization, A.F.; methodology, Y.L.; validation, Q.S.; formal analysis, Y.L. and Q.S.; investigation, Y.L. and Q.S.; resources, Y.L.; data curation, Q.S.; writing—original draft preparation, Y.L. and A.F.; Writing—review and editing, A.F.; supervision, Y.L.; project administration, Y.L.; funding acquisition, Y.L. All authors have read and agreed to the published version of the manuscript.

**Funding:** This research is funded by National Science and Technology Council: G2021172005L.

**Data Availability Statement:** The data presented in this study are available on request from the corresponding author.

**Conflicts of Interest:** The authors declare no conflict of interest.

**Appendix A**

**Table A1.** Overall reliability analysis of villagers' views in the questionnaire.

| Option | The Average Value of the Scale after Removing the Items | Scale Variance after Removing the Items | The Corrected Term and Total Correlations | Square-Wise Multiple Correlations | Cronbach $\alpha$ after the Deletion Term | Standardized $\alpha$ |
|---|---|---|---|---|---|---|
| B1.a | 110.80 | 197.520 | 0.114 | 0.717 | 0.863 | |
| B1.b | 110.88 | 192.176 | 0.391 | 0.587 | 0.858 | |
| B1.c | 111.18 | 191.840 | 0.342 | 0.611 | 0.859 | |
| B1.d | 111.28 | 189.505 | 0.304 | 0.464 | 0.860 | |
| B1.e | 110.75 | 190.799 | 0.315 | 0.627 | 0.859 | |
| B1.f | 110.97 | 183.272 | 0.515 | 0.673 | 0.854 | |
| B1.g | 111.13 | 186.870 | 0.443 | 0.644 | 0.856 | |
| B1.h | 111.00 | 189.252 | 0.421 | 0.592 | 0.857 | |
| B1.i | 111.01 | 185.402 | 0.543 | 0.649 | 0.854 | |
| B3.a | 111.23 | 189.880 | 0.266 | 0.499 | 0.862 | |
| B5.a | 111.52 | 199.400 | 0.040 | 0.336 | 0.864 | |
| B6.a | 113.03 | 184.188 | 0.460 | 0.488 | 0.855 | |
| B6.b | 112.53 | 179.878 | 0.508 | 0.473 | 0.854 | |
| C4.a | 111.77 | 189.610 | 0.292 | 0.609 | 0.861 | |
| C4.b | 111.60 | 182.999 | 0.540 | 0.646 | 0.853 | |
| C5.a | 111.48 | 187.267 | 0.325 | 0.557 | 0.860 | 0.864 |
| C5.b | 110.91 | 190.617 | 0.469 | 0.665 | 0.856 | |
| C5.c | 111.39 | 187.077 | 0.344 | 0.579 | 0.859 | |
| C5.d | 111.30 | 192.178 | 0.258 | 0.551 | 0.861 | |
| D4.a | 110.98 | 196.299 | 0.232 | 0.536 | 0.861 | |
| D5.a | 111.25 | 188.898 | 0.374 | 0.549 | 0.858 | |
| D6.a | 110.95 | 184.084 | 0.558 | 0.667 | 0.853 | |
| D6.b | 111.09 | 189.088 | 0.338 | 0.565 | 0.859 | |
| D6.c | 110.79 | 184.270 | 0.589 | 0.623 | 0.853 | |
| D6.d | 110.85 | 187.186 | 0.472 | 0.685 | 0.855 | |
| D6.e | 111.22 | 182.165 | 0.506 | 0.598 | 0.854 | |
| E8.f | 110.90 | 190.150 | 0.499 | 0.654 | 0.856 | |
| E8.g | 110.86 | 194.911 | 0.316 | 0.508 | 0.860 | |
| E8.h | 111.12 | 189.959 | 0.299 | 0.659 | 0.860 | |
| E8.i | 110.95 | 186.516 | 0.468 | 0.631 | 0.855 | |
| E8.j | 110.68 | 194.745 | 0.339 | 0.570 | 0.859 | |

Notes for Table A1:

B1: To what extent do you agree with the following statements?

a: The construction of the village's appearance is necessary for our lives.

b: The village's appearance and construction are good.

c: The village's road construction is well done.

d: The village's rivers are clean, clear, and have little pollution.

e: The efforts to improve the village's rivers are meaningful.

f: I am willing to participate in the governance of village's appearance.

g: The village's greenery and environmental protection are good.

h: The supply of water and electricity in the village is good.

i: It is necessary for the village to promote the whole appearance.

B3: If you are engaged in farming, is it convenient for you to access water for field irrigation?

B5: Do you think that installing streetlights has greatly improved the safety for nighttime travel?

B6: In the past year, how frequently have you been involved in or aware of the following activities?

a: Using the village-provided public amenities (such as the mentioned square, fitness equipment

b: Participating in public service activities for the village's appearance and construction (such as garbage cleaning and sanitation)

C4: What do you think about the following issues?

a: The village's method of used water treatment

b: The environmental harm caused by domestic used water

C5: To what extent do you agree with the following statements.

a: Used water treatment is the responsibility of the local governments.

b: Used water treatment benefits environmental development

c: The village and town promote used water treatment

d: I am very satisfied with the current domestic used water treatment methods and results

D4: How do you feel about the measure of purchasing "two-color" classified garbage bins?

D5: With the current promotion of "garbage classification," are you willing to learn about garbage classification knowledge?

D6. To what extent do you agree with the following statements?

a: The requirements for domestic waste disposal have increased

b: The implementation of garbage classification in the village is effective

c: I think garbage classification is necessary

d: I have never littered plastic bags or other garbage

e: I have started to do simple garbage sorting

E8. To what extent do you agree with the following statements?

f: It is necessary to change dry toilets to water toilets

g: I am satisfied with the cleanliness of the current household toilet

h: The construction of toilets in the village has brought convenience to me

i: I am satisfied with the cleanliness of the toilets

j: Water toilets have brought convenience to my life

## Notes

[1] The "Three-year Action" was issued by the CPC Central Committee in 2018 with the aim of promoting garbage disposal, domestic sewage treatment, toilet renovation, and village appearance in rural areas so as to construct a beautiful countryside and strengthen village planning and rural governance.

[2] The administrative village is the unit of villagers' self-governance. It is also the grassroots autonomous unit in China. The number of administrative villages is determined by the number of village committees. One administrative village may include one or more natural villages, or a large natural village can be divided into several administrative villages.

³   Remarkable results have been achieved in improving the rural living environment (People's Daily, November 12, 18th edition) (moa.gov.cn).

⁴   In China, the primary-level governments are the township-level governments.

⁵   Village committees are self-governed organizations that do not belong to governments and have no financial allocation in the formal system.

⁶   The theory of collaborative governance is a new type of management theory proposed in the 1980s. It refers to the governance between governments, enterprises, social organizations, and citizens. The core idea of collaborative governance theory is to build a more open, participatory, and co-governing political environment to improve the efficiency of governance and reduce its costs.

⁷   "Ethical bank": to establish an ethical bank account based on bank deposits, to refine and quantify the moral behavior of the villagers. It tries to manage the moral behavior of villagers in accordance with established standards, and deposit them in the ethical bank of villagers.

⁸   "points systems" are a method commonly used in rural governance. This method transforms various elements of rural governance into quantitative indicators through democratic procedures under the leadership of rural grassroots party organizations, through evaluating the daily behavior of villagers to form points and giving corresponding encouragement or rewards to form a set of effective incentives and constraint mechanisms.

⁹   The designations of "Five good villages" and "Five bad villagers" were bestowed by primary-level governments (township governments) after evaluating the following items in the village: roads sanitation, water conservancy facilities, electricity power supply, public health, and general living environments.

¹⁰  Most of the toilets in rural areas of China are waterless closets. The goal of the "toilet upgrade" is to improve toilet conditions with water closets and sewage treatment, systematically considering sewage and waste and opening up indoor sanitation possibilities. The toilet revolution can also be used to convert waste from toilets into organic fertilizer for use in farming.

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
