# Peer review of "Research on the Governance of Rural Living Environments in China: A Perspective of “System-Life” Based on Field Research Conducted in Village A, Xiangtan County, Hunan Province"

_land, doi:10.3390/land12122182_

Round 1
Reviewer 1 Report
Comments and Suggestions for Authors
I have read the manuscript with interest. This paper analyzes the tensions and interactions between local governments and social demand toward rural living environments, which has theoretical value and practical significance. It provides some insights, but the paper needs thorough changes before meeting the requirements for publication.
(1) In section 2 literature review, the manuscript introduced a large amount of specific research with the perspective of “state-society”. The conclusions of the above research might provide the reference significance, but the detailed introduction is not necessary. Meanwhile, the manuscript did not explain the essential differences between the perspective of “state-society” and “system-life”.
(2) The manuscript combined the correlation analysis of villagers’ satisfaction with environmental governance, and the differential analysis of villagers’ satisfaction with environmental policies to explore the interactive action between government and villager. However, the role of local governments in rural living environments is still difficult to reveal, the role of local governments is not only limited in environmental policies, but also include the financial investment, engineering construction, etc. Meanwhile, the role of village committees in rural living environments has not yet been explored quantitatively.
(3) The policy suggestion proposed in the manuscript are targeted, but which lack systematicity and practicality, and their guiding role in rural living environment management is limited.
(4) The conclusion is too lengthy, which makes it difficult for readers to obtain the main findings and insights of this study.
(5) The manuscript lacks the research Frontiers in relevant fields, a large number of references are Chinese references.
(6) The manuscript needs more in-depth improvements in English. Some of the language expressions are not accurate enough, it should be more concise and accurate.
Comments on the Quality of English LanguageI have read the manuscript with interest. This paper analyzes the tensions and interactions between local governments and social demand toward rural living environments, which has theoretical value and practical significance. It provides some insights, but the paper needs thorough changes before meeting the requirements for publication.
(1) In section 2 literature review, the manuscript introduced a large amount of specific research with the perspective of “state-society”. The conclusions of the above research might provide the reference significance, but the detailed introduction is not necessary. Meanwhile, the manuscript did not explain the essential differences between the perspective of “state-society” and “system-life”.
(2) The manuscript combined the correlation analysis of villagers’ satisfaction with environmental governance, and the differential analysis of villagers’ satisfaction with environmental policies to explore the interactive action between government and villager. However, the role of local governments in rural living environments is still difficult to reveal, the role of local governments is not only limited in environmental policies, but also include the financial investment, engineering construction, etc. Meanwhile, the role of village committees in rural living environments has not yet been explored quantitatively.
(3) The policy suggestion proposed in the manuscript are targeted, but which lack systematicity and practicality, and their guiding role in rural living environment management is limited.
(4) The conclusion is too lengthy, which makes it difficult for readers to obtain the main findings and insights of this study.
(5) The manuscript lacks the research Frontiers in relevant fields, a large number of references are Chinese references.
(6) The manuscript needs more in-depth improvements in English. Some of the language expressions are not accurate enough, it should be more concise and accurate.
Author Response
Reviewer 1.
We wish to thank you for your comments and suggestions. They have been very helpful in clarifying and strengthening the manuscript. We have focused our attention on the Literature review, the research question and the Conclusions.
Question 1:
In section 2 literature review, the manuscript introduced a large amount of specific research with the perspective of “state-society”. The conclusions of the above research might provide the reference significance, but the detailed introduction is not necessary. Meanwhile, the manuscript did not explain the essential differences between the perspective of “state-society” and “system-life”.
Response 1. We have scaled down the literature review and added some international references to link to the global discourse on rural governance.
Response 2. According to your helpful suggestion, we have explained more clearly the essential differences between the perspectives of “State-Society” and “System-Life”.
2.1 From the perspective of “State-Society”, governments take measures to fulfill administrative tasks aimed at improving the living environment in rural areas. These measures include refining relevant legislation, implementing policies utilizing, and employing modern governance technologies such as information networks. We have reduced this explanation in the text.
2.2 The “System-Life” perspective incorporates the more subjective views of the villagers and social organizations in rural governance. Authorities play a supportive role to activate the potential of the villagers. It is confirmed that governments should pay much more attention to the views of villagers. (also In Conclusions).
Question 2. The manuscript combined the correlation analysis of villagers’ satisfaction with environmental governance, and the differential analysis of villagers’ satisfaction with environmental policies to explore the interactive action between government and villager. However, the role of local governments in rural living environments is still difficult to reveal, the role of local governments is not only limited in environmental policies, but also include the financial investment, engineering construction, etc. Meanwhile, the role of village committees in rural living environments has not yet been explored quantitatively.
Responses:
2.1 The role of local governments in improving rural living environments is difficult to fully determine as it requires interdepartmental contacts.
2.2 From the perspective of “System-Life”, we explain that most of the roles and functions of governments are supportive roles.
The paper concludes that the village committee is the bridge connecting the primary-level government and the village people. We plan to explore quantitatively the role of the village committee in rural living environments in the future.
Question 3. The policy suggestions you proposed in the manuscript are targeted, but which lack systematicity and practicality, and their guiding role in rural living environment management is limited.
Response: As you observe, the policy suggestions proposed in the manuscript are targeted, to some extent. In reality such directions are practical, because village people pay attention to the policies that benefit them. For example, “ Toilet Revolution”, but the subsidy offered by the local governments is limited.
Question 4. The conclusion is too lengthy, which makes it difficult for readers to obtain the main findings and insights of this study.
Response: The conclusion has been narrowed down. Two paragraphs were removed from the text and many words and phrases deleted. In total over 540 words have been deleted, but are not shown in the text.
Question 5: The manuscript lacks the research Frontiers in relevant fields, a large number of references are Chinese references.
Response: Some references to foreign experts were added.
Question 6: The manuscript needs more in-depth improvements in English. Some of the language expressions are not accurate enough, it should be more concise and accurate.
Response: We have checked the manuscript and made many improvements in English.
Reviewer 2 Report
Comments and Suggestions for Authors
· It is suggested to formulate a more concrete research aim as it is not really seen in the Abstract, also nor in Introduction part.
· What is the role of „bottom up approach“ living in the “System-Life”?
· Not very clear what was asked in the interview, how many people participated in the interview, their roles in „System-Life“? Interview results could be presented in more precise way – e. g. giving some table, etc.
· Technical revision of text should be done – text shrift etc.
· What other China villages could learn from this article – practical and scientific insights?
Author Response
Many thanks for the comments and suggestions of Reviewer 2. They have been very helpful in clarifying the manuscript.
Question 1. It is suggested to formulate a more concrete research aim as it is not really seen in the Abstract, also nor in Introduction part.
Response: The research question was originally found at the end of the literature review, now it has been removed to the Introduction part.
Question 2. What is the role of “bottom up approach” in the“System-Life”?
Response: The bottom-up approach in “System-Life” could identify the real needs of the villagers, so that local governments could formulate policies respecting the living habits and real ideas of the villagers. It can reduce the possible conflict between the government and the village people. As well, it could help villagers to be more willing to participate in these governance activities
Question 3. Not very clear what was asked in the interview, how many people participated in the interview, their roles in “System-Life”? Interview results could be presented in more precise way – e. g. giving some table, etc.
Responses:
3.1 There are 302 villagers who participated in the survey. (P7) Their roles are different based on their attitudes, some of them do not care about the policies for improving rural living environments which is not related to their needs, many are also reluctant to participate in governance activities. While some villagers would like to participate in governance policies, such as “Toilet Revolution” which would partially meet their needs.
3.2 The table that reflects “the content and result of interview” was supplemented, please refer to Table 3 in the Annex.
Question 4.
Technical revision of text should be done- text shrift etc.
Response: The whole text and the tables have been checked and adjusted.
Question 5: What other China villages could learn from this article–practical and scientific insights?
Response: In China, villages can be classified into different categories according to their environmental characteristics. In the paper, it was shown that Village A was rated among the “five good villages” in the fourth quarter of 2019. However, there were still some problems, such as the low participation of the villagers; strong preference of the villagers for the “water closet” while the subsidy of toilet for per household is not sufficient, etc. Other villages in China could learn from this article that it is important to identify the real needs of the villagers, and find the common interest between the local governments, village committees and village people.
Reviewer 3 Report
Comments and Suggestions for Authors
Dear authors and editor,
The manuscript is interesting and contemporary. The problems of many countries in the world, certainly rich countries in certain parts of their territory have similar problems. The abstract is complete. It contains all relevant facts from issues, methods to results. However, improving just a few more parts will make the manuscript better.
- In line 104, the authors say "Based on relevant theories such as collaborative governance...". Explain in a footnote “theory collaborative governance”.
- Figure 1 is explained in one sentence in lines 166 and 167. This scheme needs to be explained in more detail. It is not clear why there are circles of different sizes on the right, arrows going in both directions. So, it is necessary to explain in detail and connect it with the problem that is investigated in the paper.
- In the questionnaire there was a question about ethnicity. Is that a necessary question? Would there be a difference in the responses if ethnicity was heterogeneous? And so it is not analyzed at the level of ethnicity.
- In the line 287 it says “..null hypothesis is rejected ...”. Where the hypotheses are stated in the paper, including the null hypothesis, so that you could have rejected it.
- In Table 3, what does it mean in column Option B1.a, B1.b, B1.c .... E8.j?
- For values expressed in Chinese yuan, put values in euros or dollars in parentheses.
- In an interview, it's an interesting statement that should have been analyzed, because of the government's desire to separate garbage, and then they don't do it. What kind of satisfaction is that for the peasants? It may be a paper in itself, but it deserves a little more attention here. You looked back in the conclusion, but it should also be in the analysis.
- In lines In lines 550 and 551 the terms are listed “Demonstration Village” and “Star Village”, “...which means various extra resources were allocated by the primary-level government...” What are the extra resources?
An explanation of the above will further clarify the manuscript and thus make it better.
Best regards
Author Response
Reviewer 3
Many thanks for your comments and positive suggestions. They have been very helpful in clarifying the manuscript.
Question1: In line 104, the authors say "Based on relevant theories such as collaborative governance...". Explain in a footnote “theory collaborative governance”.
Response: A footnote on the“theory of collaborative governance” was added. (P4)
Question 2: Figure1 is explained in one sentence in lines 166 and 167. This scheme needs to be explained in more detail. It is not clear why there are circles of different sizes on the right, arrows going in both directions. So, it is necessary to explain in detail and connect it with the problem that is investigated in the paper.
Response: The explanation of Figure1 was supplemented under the Figure. (P6)
Question 3: In the questionnaire there was a question about ethnicity. Is that a necessary question? Would there be a difference in the responses if ethnicity was heterogeneous? And so it is not analyzed at the level of ethnicity.
Response: “Ethnicity” is not a necessary factor, and it was removed from the whole text and Table 1.
Question 4: In line 287 it says “..null hypothesis is rejected ...”. Where the hypotheses are stated in the paper, including the null hypothesis, so that you could have rejected it.
Response: “The null hypothesis is rejected…” , this sentence was deleted in the text (P9).
Question 5: In Table 3, what does it mean in column Option B1.a, B1.b, B1.c .... E8.j?
Response: B1. a, etc refers to selected questions, please see the details under Table 3 in the annex, we listed them as notes. (P21-22)
Question 6: For values expressed in Chinese yuan, put values in euros or dollars in parentheses.
Response: The values of Chinese yuan are indicated in US dollars in parentheses (P9-10)
Question 7: In an interview, it's an interesting statement that should have been analyzed, because of the government's desire to separate garbage, and then they don't do it. What kind of satisfaction is that for the peasants? It may be a paper in itself, but it deserves a little more attention here. You looked back in the conclusion, but it should also be in the analysis.
Response: The villagers are mostly satisfied with garbage classification, not because of the garbage classification itself, but because finally there was a reconciliation between the local governments and villagers. That is, the villagers have the right to deal with the kitchen waste and dead leaves to compost and reuse by themselves, other large plastic packaging and waste is required to be put into the garbage cans. In that case, the villagers could benefit from dealing with the garbage. (P12)
Question 8. In lines 550 and 551 the terms are listed “Demonstration Village” and “Star Village”, ...which means various extra resources were allocated by the primary-level government...” What are the extra resources?
Response: As for the “Demonstration Village” and “Star Village”, the extra resources usually refer to increased financial resources allocated by the higher-level governments, to ensure that the “demonstration village” or “Star village” is built sustainably.
Round 2
Reviewer 1 Report
Comments and Suggestions for Authors
The article was clarified at important points and necessary references were added. However, the conclusions still is too length and the References should added more international research achievements in related fields.
Comments on the Quality of English LanguageThe quality of English Language basically meets publishing requirements, but some language expressions need to be more concise and accurate.
